# Understanding Somatic Symptoms Associated with South Korean Adolescent Suicidal Ideation, Depression, and Social Anxiety

**DOI:** 10.3390/bs11110151

**Published:** 2021-11-01

**Authors:** Hayoung Kim Donnelly, Danielle Richardson, Scott V. Solberg

**Affiliations:** Department of Counseling Psychology and Applied Human Development, Wheelock College of Education and Human Development, Boston University, Boston, MA 02215, USA; hyjkim@bu.edu (H.K.D.); drich@bu.edu (D.R.)

**Keywords:** suicide, depression, anxiety, somatic symptoms, Korean youth

## Abstract

Korea’s suicide rate has increased steadily in recent years and it has become the leading cause of death among Korean youth. This paper aims to propose suicide risk identification strategies for South Korean youth by identifying profiles of suicide risk alongside reported somatic complaints. For several reasons, somatic complaints are more commonly reported than mental health concerns in Korea, where somatic complaints are likely to be representative of larger mental health worries. Nationally representative data of Korean first-year middle school students were used to identify mental health profiles by examining reported suicidal ideation, depression, and social anxiety and the prediction effect of reported somatic symptoms within these profiles. Results indicated that female students reported a greater level of suicidal ideation, depression, and social anxiety compared to male students. Each gender (females and males) exhibited five different mental health profile groups, which ranged from low risk to high risk. Somatic symptoms (sleep, stomach ache, tiredness, breathing, appetite, headache, fever, nausea) significantly predicted each profile group, with sleep issues serving as the strongest predictor for risk across both genders and all groups. Therefore, for mental health professionals working with Korean youth, it is encouraged to identify and recognize somatic complaints as potentially representative of mental health concerns and suicidality risk.

## 1. Introduction

In his 2010 Presidential Address, John Westefeld implored counseling psychologists to increase engagement in suicide research and prevention and, more recently, has argued that efforts to address suicide rates are a significant social justice concern [1,2]. In the United States, 14.2 per 100,000 individuals die from suicide each year [3,4]. Approximately 800,000 individuals worldwide die from suicide each year, with around 60% of these deaths occurring in Asia [5]. This paper is specifically focused on suicide rates in South Korea (Korea, hereafter), which has the 10th overall highest suicide rate in the world and ranks highest among all Asian and economically developed countries [3,6]. Moreover, while rates are decreasing in many countries, Korea’s suicide rate has increased steadily since 1995 [7] and it is now estimated at approximately 27 deaths per 100,000 people [3,4]. Among Korean adolescents and emerging adults, suicide is the leading cause of death, representing 44.8% of all deaths [8]. In addition, suicidal ideation is a proximal risk factor for suicide attempts and deaths. Rates of suicidal ideation are greater in female youth [9] while rates of suicide deaths are greater in male youth [10]. 

Considering the significance of suicide and suicidal ideation in Korean youth, the main objective of this paper’s research is aimed at improving risk identification. Thus far, utilization of psychotherapy treatment in Korea is noticeably low. A 2006 study sampling 3717 Korean citizens found that only 15.7% of the sample had consulted a physician within the past year [11]. Participants with major depressive disorder, however, were more likely to consult a physician within the year (31.9% of the sample). However, only 2% of Koreans diagnosed with major depressive disorder cited depressive symptomatology as a motive for the healthcare visit [11]. There is evidence that Koreans—and, by extension, Korean youth—who are suffering from mental health challenges may be more likely to report physical and somatic complaints rather than mental health concerns [12,13,14]. This reporting pattern is also present regarding suicidality. When examining suicidality in Korean adolescents, psychosomatic symptoms are a significant risk factor for attempted suicide [15]. Additionally, clinical characteristics of 391 suicide deaths from The Korea Psychological Autopsy Center reported that 48.2% of the sample signaled somatic complaints before dying by suicide [16]. Further, this report noted several significant behavioral signals in this sample before suicide, such as changes in sleep (65.9%), changes in appetite (53.4%), and difficulty concentrating (32.9%). The literature also suggests that reported somatic complaints differ by gender in youth [17]. The unique pattern for males and females can be understood within the context of development, with female youth experiencing pubertal maturity, which in turn provokes different physiological and psychological changes than males, therefore influencing reported mental and somatic complaints [18]. Overall, these somatic concerns have now been understood to be symptoms and reflective of mental health concerns, though they are not perceived as such. 

Societal stigma about seeking help for mental health concerns may contribute to Korea’s high suicidality rate, less disclosure of suicidal ideation, but greater reporting of somatic symptoms [19,20]. Stigma has been shown to impede help-seeking behaviors regarding mental health concerns [21,22,23], especially among Asians and Asian Americans [24,25]. The effects of stigma on mental health and access to care have been observed commonly across many countries, regardless of the countries’ individualist or collectivist affiliation. Although stigmatization of mental illness is present within Western countries, the discrimination in a family-oriented collectivistic society can be more severe [19] and impede help-seeking behaviors. In Asian cultures, this is in part due to the belief that the stigma of mental illness reflects negatively on the entire family [19,26]. 

In addition to the effects of a collectivist culture, the perfectionist culture of Korea has impacted the low disclosure rate of suicidal ideation within Korean society. Generally, perfectionism shapes the binary thinking of individuals about their lives and enforces a “success or failure” dichotomy [27]. Especially within the view of labeling theory, youth who are given “negative” labeling are more likely to accept their self-identity as a “socially failed” or “useless member of society” [28]. Considering that Korean society has traditionally emphasized the success of the community rather than the growth of the individual, experiencing or disclosing suicidal ideation may have negative consequences. A person experiencing these mental health concerns may be viewed negatively, from themselves or from society. In other words, Korean youth may benefit from safe spaces where their suicidal thoughts can be perceived and treated as just one of the stories in their life rather than the conclusion of their whole life. 

Due to rapid socio-economic and industrial changes in recent decades [29], there has been a rise in the number of Korean individuals who present with a clinically significant mental health concern, such as depression, suicidal ideation, and suicidality [29,30,31]. The Korean government responded to this rise by establishing the Mental Health Act in 1995, which mandated local governments to provide mental health services through already existing public health centers, with the goal to increase community mental health services [32]. This shift to a community prevention strategy was new to Korea, which has historically focused on inpatient and hospitalization treatment for mental health concerns [32]. Since the introduction of the Mental Health Act, Korea has seen an approximately five-fold increase in the number of mental health rehabilitation centers [33]. This act encouraged the spending portion of the government’s budget to fund the creation and care of community mental health centers. The WHO reports that 6% of Korea’s healthcare expenditure is assigned to mental health, with approximately 31% assigned to mental hospitals [34]. However, Korea’s current mental health system is still largely framed by the hospitalization model as opposed to a community-based approach [33]. To combat the more recent rising rates of suicide, Korea’s Ministry of Health and Welfare announced plans to reduce the suicide rate to less than 20 per 100,000 people by the year 2022 through their National Suicide Prevention Action Plan [35]. To accomplish this, Korea’s government created “Suicide CARE”, which is a standardized suicide prevention program for gatekeeper intervention. Suicide CARE was recently revised between August 2019 and February 2020, and as of 2019, 1.2 million individuals have completed this program [36]. 

Despite increased efforts to improve mental health interventions, the low rates of disclosure and help-seeking behaviors regarding mental health among Koreans persist. For example, while 25.4% of Koreans experience at least one mental health concern (such as suicidal ideation, depression, anxiety, or substance use), less than 10% report reaching out to psychiatrists, mental health specialists, or religious leaders for mental health services [37]. To support community health specialists and educators in identifying suicidality and mental health concerns in Korean youth, physical and psychosomatic complaints should be examined as representative of suicidality and mental health concerns (such as depression and social anxiety). It would be beneficial to highlight and identify specific physical and psychosomatic complaints related to mental illness in order to better inform treatment professionals and prevent long-term chronic illness. Somatic symptoms are considered a health issue that, due to labeling and stigma, are easier to disclose. Korean individuals tend to be more sensitive or report more frequently their somatic well-being compared to reporting their emotional well-being. 

Up to now, the relation between suicidality and somatic symptoms has been understudied with Korean youth. Many psychological studies examining suicide and physical health in Korea have been focused on the relationship between chronic diseases such as asthma, heart disease, and diabetes rather than somatic symptoms occurring in daily life, such as sleep and appetite. Among the few studies considering common somatic symptoms, the primary focus is on Korean children or adult populations [38]. These results are difficult to generalize to the general population across different developmental phases, and far too little attention has been paid to the youth of Korea on this topic. Therefore, this study aims to fill the research gap of specific somatic risk factors for mental health concerns in Korean youth. Following recommendations from Wong, Maffini, and Shin [39], this study seeks to identify culturally relevant screening indicators that psychologists can use to train and orient community healthcare professionals and educators to assess for suicidal risk among Korean youth. 

### Current Study

Using Korean national data on youth, this study sought to evaluate whether a screening system could be established that identifies profiles of suicide risk that correlate with somatic symptoms. Specifically, the current study examined key mental health indicators of suicidal ideation, depression, and social anxiety in relation to reported somatic symptomology. By considering not only suicidal ideation but also depression and social anxiety, the study sought to create a multi-dimensional picture of Korean youth’s mental health. Furthermore, by demonstrating a prediction effect of somatic symptoms within different profiles of mental health, this study aimed to provide the Korean community of healthcare providers and educators with screening strategies that will alert them to somatic symptoms reported among youth that may indicate potential suicidal ideation and risk.

## 2. Method

### 2.1. Sample

The secondary data used in this study were drawn from the “Korea Children and Youth Panel Survey 2018 (KCYPS)”, which were originally collected by the National Youth Policy Institute (NYPI) and are openly available on the NYPI data archive (https://www.nypi.re.kr) (accessed on 11 November 2020). The secondary data were a nationally representative sample of Korean youth who were in their first year of middle school (14 years old). Multi-stage stratified cluster sampling was used with the consideration of the proportion of students across 16 administrative districts of South Korea. Data collection was conducted in person by trained data collectors and youth were asked to respond using tablet computers. For this study, the sample consisted of the original KCYPS 2018 data for 2590 youth. There were no missing values across all variables and youth respondents. A total of 54.2% of the students identified as male and 45.8% identified as female.

### 2.2. Measures

Suicidal ideation, depression, social anxiety, and somatic symptoms were used in this study. Youth were asked to respond how much they agreed with the given sentences related to suicidal ideation, depression, and social anxiety (1 = Strongly disagree, 2 = Disagree, 3 = Agree, 4 = Strongly agree). Suicidal ideation was measured with one item asking “I have a thought that I want to die” [40]. Depression was measured with nine items with the reliability of 0.91 (e.g., “I have lots of worries”, “I feel lonely”, and “I have a lack of interest in everything”) [40]. Social anxiety was measured with five items with the reliability of 0.87 (e.g., “It is hard to say clearly about my opinion to other people”, “I don’t like to be in front of people”) [41].

Somatic symptoms related to sleep, tiredness, appetite, nausea, breathing, fever, and headache were measured from the questions developed by Cho and Lim [42]. The youth responded how much they agree with the given sentences related to each somatic symptom (1 = Strongly disagree, 2 = Disagree, 3 = Agree, 4 = Strongly agree). Sleep was measured with the statement of “I can’t sleep deeply and I used to wake up”, tiredness was measured with “I often feel tired”, appetite was measured with “Sometimes I have no appetite”, nausea was measured with “I often feel sick at the stomach”, breathing was measured with “Sometimes it is hard to breathe”, fever was measured with “I often feel like having a fever”, and headache was measured with “I often have headache”. The standardized values of suicidal ideation, depression, social anxiety, and somatic symptoms were used for the analyses.

### 2.3. Plan for Analyses 

#### 2.3.1. Primary Analysis Using Latent Profile Analysis

Different latent profiles of suicidal ideation, depression, and social anxiety among Korean youth were identified using Latent Profile Analysis (LPA) with Mplus 7.0 [43]. LPA is a response-centered clustering method. Unlike variable-centered clustering methods, LPA clusters each individual based on the similarity and differences within the overall profile across responses, rather than the level of responses. The variable-centered approach identifies clusters by certain standards or thresholds, such as classifying each individual by clinical guidelines or quartiles. However, the response-centered approach focuses on identifying clusters by the relative deviation in profiles within the data. Each cluster is labeled by comparing its pattern with patterns of other clusters. By examining the number of latent profiles and their profiles across variables, the LPA analysis was used to identify whether three reported mental health concerns among Korean youth, namely suicidality, depression, and social anxiety, can be clustered based on their profiles. Indices used for determining the final model that includes the optimal number of profiles that emerge from the data include: Akaike Information Criterion (AIC), Bayesian Information Criterion (BIC), Sample-Size Adjusted Bayesian Information Criterion (SSABIC), entropy, and Adjusted Lo–Mendell–Rubin likelihood ratio test (Adj.LRT). AIC, BIC, and SSABIC provide information regarding how much each model with a different number of profiles fits in explaining the data. A model with the lowest scores in three indices is considered as the optimal model [44]. Entropy explains the clustering quality of a model. A model with the highest score in entropy is considered the optimal model [45]. Adj. LRT compares whether there are significant differences between a model with N profiles (e.g., five-profile model) and a model with N-1 profiles (e.g., four-profile model) [46]. If there is no statistically significant difference between these criteria between the two models, we assume that distributing samples to create one more class is not necessary. In other words, if the *p*-value of class N is higher than 0.05, this means that the model with N profiles is not significantly different from the model with N-1 profiles, and it is recommended to select a model with N-1 profiles rather than N profiles. 

#### 2.3.2. Post-Hoc Analysis Using Logistic Regression

As a follow-up to the main results, post-hoc analyses were conducted using binary logistic regression in order to determine whether specific somatic symptoms were able to distinguish between different latent profiles for suicidal ideation, depression, and social anxiety. After identifying different latent groups, our research team planned additional analysis to provide practical information in using and predicting latent groups of mental health into practice. As discussed in the literature, Korean youth tend to share their somatic symptoms more openly than mental health issues. Therefore, we especially focused on somatic symptoms in follow-up analysis to predict latent groups. For this purpose, logistic regression was fitted to examine the odds of prediction across different latent groups, which were categorical dependent variables, by using different somatic symptoms, which were continuous independent variables. In other words, youth considered high-risk or low-risk were compared using somatic symptoms as independent variables and the latent mental health profiles as dependent variables.

## 3. Results

### 3.1. Prevalence Rates among Korean Youth

Before conducting the primary analyses, a one-way MANOVA was conducted to assess for potential gender differences for reported suicidal ideation, depression, social anxiety, and physical symptoms. The results showed that female students (*n* = 1185) reported higher ratings for all the mental health and symptom variables than male students (*n* = 1405).

Korean female students (M = 1.70, SD = 0.81) reported significantly higher levels of suicidal ideation than male students (M = 1.45, SD = 0.68) (F = 73.608, *p* < 0.001, *η*^2^ = 0.028). Female students (M = 1.94, SD = 0.65) reported significantly higher levels of depression than male students (M = 1.73, SD = 0.63) (F = 72.848, *p* < 0.001, *η*^2^ = 0.027). Female students (M = 2.23, SD = 0.76) reported significantly higher levels of social anxiety than male students (M = 2.09, SD =0.74) (F = 24.797, *p* < 0.001, *η*^2^ = 0.009). Aligned results were identified in the comparison of somatic symptoms between female students and male students. Female students (M = 1.85, SD = 0.88) reported higher levels of sleep issues than male students (M = 1.74, SD = 0.82) (F = 10.222, *p* < 0.01, *η*^2^ = 0.004). Female students (M = 2.50, SD = 0.97) reported a higher level of tiredness than male students (M = 2.21, SD = 0.95) (F = 59.598, *p* < 0.001, *η*^2^ = 0.023). Female students (M = 2.16, SD = 0.93) reported more issues in losing appetite than male students (M = 2.02, SD = 0.89) (F = 15.761, *p* < 0.001, *η*^2^ = 0.006). Female students (M = 1.86, SD = 0.86) reported more nausea issues than male students (M = 1.67, SD = 0.76). (F = 32.694, *p* < 0.001, *η*^2^ = 0.012). Female students (M = 1.71, SD = 0.79) reported more issues in breathing than male students (M = 1.59, SD = 0.74) (F = 15.369, *p* < 0.001, *η*^2^ = 0.006). Female students (M = 1.79, SD = 0.85) reported higher levels of fever issues than male students (M = 1.65, SD = 0.77), (F = 20.370, *p* < 0.001, *η*^2^ = 0.008). Female students (M = 2.01, SD = 0.91) reported more headaches than male students (M = 1.82, SD = 0.83) (F = 29.027, *p* < 0.001, *η*^2^ = 0.011). Female students (M = 1.71, SD = 0.79) reported more stomach ache than male students (M = 1.62, SD = 0.72) (F = 10.509, *p* < 0.01, *η*^2^ = 0.004).

### 3.2. Latent Mental Health Profiles among Korean Youth 

As with the results of the MANOVA above, female and male students consistently showed different prevalence rates in suicidal ideation, depression, and social anxiety. Therefore, we focused on finding mental health profiles by different genders of students rather than all students at once. For both female and male students, multiple model fit criteria—AIC, BIC, and SSABIC (information criterion), entropy (classification quality), and Adj.LRT (likelihood ratio statistical test)—were presented to identify the number of profiles across levels of suicidal ideation, depression, and social anxiety (Table 1). As the result of the model fit comparison, each five-profile model was selected for both the male and female groups. AIC, BIC, and SSABIC consistently decreased when the number of profiles increased. The entropy of each three-profile model of male and female was the highest compared to other models. However, the results of Adj. LRT showed that the four-profile model of male students was significantly different from the three-profile model of male students, and the five-profile model of male students was also significantly different from the four-profile model of male students. However, the six-profile model of male students was not significantly different from the five-profile model. The same result was identified in the female groups. The results of Adj. LRT showed that the four-profile model of female students was significantly different from the three-profile model of female students, and the five-profile model of female students was also significantly different from the four-profile model of female students, but the six-profile model of female students was not significantly different from the five-profile model of female students. Considering the overall results across different model fit criteria, the five-profile models from male and female students were deemed to be the best-fitted models for describing diverse profiles across suicidal ideation, depression, and social anxiety for each male and female group. 

For male students, Figure 1 shows the five different profiles identified by Latent Profile Analysis. Each group is named based on the profile across suicidal ideation, depression, and anxiety: *high-risk group, moderate-risk group 1, moderate-risk group 2, mild-risk group, low-risk group*. The first group consisting of 7% of the male youth population is referred to as the high-risk group due to their having the highest reported levels of suicidal ideation (m = 2.22). This group also reported higher levels of depression (m = 1.60) and anxiety (m = 0.63) compared to the mean of all students (m = 0). The second group consisting of 3% of male youth is referred to as moderate-risk group 1 due to the level of suicidal ideation (m = 0.58) being moderately higher than average, and the level of depression (m = 1.11) and anxiety (m = 1.21) is higher than average. The third group consisted of 26% of male youth and is referred to as moderate-risk group 2 due to reported moderate but higher levels of suicidal ideation (m = 0.58), depression (m = 0.33), and anxiety (m = 0.11) than average. The fourth group consisting of 27% of the male youth is referred to as the mild-risk group based on reporting some elevation in ratings for anxiety (m = 0.59) but the ratings for suicidal ideation (m = −0.75) and depression (m = −0.04) being lower than average. The final group consisted of 37% of male youth and is referred to as the low-risk group due to reporting low levels across all three mental health areas—suicidal ideation (m = −0.75), depression (m = −1.02), and social anxiety (m = −0.99).

For female students, Figure 2 shows the different profiles in five groups derived by Latent Profile Analysis. As with the profiles of male students, each group of female students is named based on the profile across suicidal ideation, depression, and anxiety: high-risk group, moderate-risk group 1, moderate-risk group 2, mild-risk group, low-risk group. The first group, consisting of 16% of the female youth population, is referred to as the high-risk group due to their having the highest reported levels of suicidal ideation (m = 2.16). This group also reported higher levels of depression (m = 1.28) and anxiety (m = 0.59) compared to the mean of all students (m = 0). The second group, consisting of 34% of female youth, is referred to as moderate-risk group 1 due to the level of suicidal ideation (m = 0.58) being moderately higher than average. This group also reports levels of depression (m = 0.57) and anxiety (m = 0.30) that are moderately higher than average. The third group consisted of 3% of female youth and is referred to as moderate-risk group 2 due to reported higher levels of depression (m = 1.24) and anxiety (m = 1.20) but without suicidal ideation (m = −0.75). The fourth group, consisting of 19% of the female youth, is referred to as the mild-risk group based on reporting some elevation in ratings for anxiety (m = 0.51) but the ratings for suicidal ideation (m = −0.75) and depression (m = −0.20) being lower than average. The final group consisted of 27% of female youth and is referred to as the low-risk group due to reporting low levels across all three mental health areas—suicidal ideation (m = −0.75), depression (m = −0.81), and social anxiety (m = −0.82).

### 3.3. Somatic Symptoms Associated with Mental Health Profiles 

For suggesting the clinical application in predicting different mental health profiles of youth, this study conducted logistic regression as a post-hoc test to assess for differences between the groups generated from the LPA. Figure 3 provides four different prediction steps outlined to distinguish between male youth with higher risk and male youth with lower risk. As the first step, somatic symptoms were used to distinguish male youth in the risk groups (high-risk group, moderate-risk group 1, moderate-risk group 2, mild-risk group) and male youth in the low-risk group. In the second step, among male youth in the four risk groups, somatic symptoms were used to predict whether male students have higher suicidal ideation than average (high-risk group, moderate-risk group 1, moderate-risk group 2) or lower suicidal ideation than average (mild-risk group). In Step 3, between groups with suicidal ideation (high-risk group, moderate-risk group 1, moderate-risk group 2), somatic symptoms were used to predict whether the level of suicidal ideation was relatively higher (high-risk group) or lower (moderate-risk group 1 and 2). As Step 4, between the two emerging but relatively lower suicidal ideation groups (moderate-risk group 1 and 2), somatic symptoms were used to predict whether the level of depression and anxiety was relatively higher (moderate-risk group 1) or lower (moderate-risk group 2).

For female students, Figure 4 provides four different prediction steps outlined to distinguish between female youth with higher risk and female youth with lower risk. In Step 1, somatic symptoms were used to distinguish female youth in the risk groups (high-risk group, moderate-risk group 1, moderate-risk group 2, mild-risk group) and female youth in the low-risk group. In Step 2, among female youth in the four risk groups, somatic symptoms were used to predict whether female students had higher suicidal ideation than average (high-risk group, moderate-risk group 1) or lower suicidal ideation than average (moderate-risk group 2, mild-risk group). In Step 3, between groups with suicidal ideation (high-risk group, moderate-risk group 1), somatic symptoms were used to predict whether the level of suicidal ideation was relatively higher (high-risk group) or lower (moderate-risk group 1). In Step 4, between the two groups that reported higher depression and anxiety but lower suicidal ideation than average (moderate-risk group 2, mild-risk group), somatic symptoms were used to predict whether the level of depression and anxiety was relatively higher (moderate-risk group 2) or lower (mild-risk group).

As the results of examining the prediction effect of somatic symptoms in each step using logistic regression (Table 2), eight somatic symptoms (sleep, stomach ache, tiredness, breathing, appetite, headache, fever, nausea) predicted at least one step from either male or female youth. Among all symptoms, sleep was a statistically significant predictor in most of the steps, with three steps in male youth (Step 1, 2, and 4) and three steps in female youth (Step 1, 2, and 3). Male students having difficulty with sleep were more likely to be in the high-risk groups than low-risk groups (Step 1: *β* = 0.25, *p* < 0.01; Step 2: *β* = 0.18, *p* < 0.05, Step 4: *β* = 0.63, *p* < 0.01). Female students having difficulty with sleep were more likely to be in the high-risk groups than low-risk groups (Step 1: *β* = 0.36, *p* < 0.01; Step 2: *β* = 0.28, *p* < 0.01, Step 3: *β* = 0.23, *p* < 0.05). 

The physical symptoms, i.e., stomach ache, tiredness, and breathing, showed the second-best predictive effect in screening following symptoms with sleep. Stomach ache was a statistically significant predictor in two steps in male youth (Step 1 and 2) and two steps in female youth (Step 1 and 4). Male students who experienced stomach ache more often were more likely to be in the high-risk groups than low-risk groups (Step 1: *β* = 0.21, *p* < 0.05; Step 2: *β* = 0.23, *p* < 0.05). Female students who experienced stomach ache more often were more likely to be in the high-risk groups than low-risk groups (Step 1: *β* = 0.25, *p* < 0.01; Step 4: *β* = 0.55, *p* < 0.01). Tiredness was a statistically significant predictor in two steps in male youth (Step 1 and 4) and two steps in female youth (Step 1 and 3). Male students who felt tiredness more often were more likely to be in the high-risk groups than low-risk groups (Step 1: *β* = 0.37, *p* < 0.001; Step 4: *β* = 0.98, *p* < 0.01). Female students who felt tiredness more often were more likely to be in the high-risk groups than low-risk groups (Step 1: *β* = 0.30, *p* < 0.01; Step 3: *β* = 0.30, *p* < 0.05). Breathing was a statistically significant predictor in two steps in male youth (Step 1 and 2) and two steps in female youth (Step 1 and 2). Male students who experienced breathing problems more often were more likely to be in the high-risk groups than low-risk groups (Step 1: *β* = 0.43, *p* < 0.001; Step 2: *β* = 0.22, *p* < 0.05). Female students who experienced breathing problems more often were more likely to be in the high-risk groups than low-risk groups (Step 1: *β* = 0.38, *p* < 0.01; Step 2: *β* = 0.26, *p* < 0.01).

Appetite had a predictive effect in only one step for both male and female youth. Male students who lost their appetite were more likely to be in the high-risk groups than low-risk groups (Step 1: *β* = 0.28, *p* < 0.001). Female students who lost their appetite were more likely to be in the high-risk groups than low-risk groups (Step 1: *β* = 0.30, *p* < 0.01). Headache, fever, and nausea had predictive effects in only one step. Male students who had headache (Step 4: *β* = 0.92, *p* < 0.01) or fever (Step 1: *β* = 0.86, *p* < 0.001) were more likely to be in the high-risk groups than low-risk groups. Female students who experienced nausea were more likely to be in the high-risk groups than low-risk groups (Step 1: *β* = 0.37, *p* < 0.001).

## 4. Discussion

This study looked at a nationally representative sample and examined patterns and somatic symptomatology predictors for suicidality and mental health in Korean youth. Post-hoc analyses were used to determine if there were key clinical factors that mental health professionals could use to identify and subsequently treat at-risk youth. Among Korean youth, the patterns across suicidal ideation, depression, and social anxiety were not identical, both across different gendered students and also across students within the same gender. Each gender had five different patterns across suicidal ideation, depression, and social anxiety. There were four risk groups and one low-risk group for each gender. Within the four risk groups, there were students who reported depression and social anxiety with suicidal ideation but there were also students who reported depression and social anxiety without suicidal ideation. Using logistic regression for post-hoc analysis, all somatic symptoms—sleep, stomach ache, tiredness, breathing, appetite, headache, fever, nausea—significantly predicted one or more steps in screening within either male or female youth (to see each step in the screening process, please see Figure 3). 

When comparing rates of reported mental health concerns and somatic complaints, several significant gender differences were identified. Consistent with the previous literature [9,47], female Korean youth reported a significantly higher proportion of suicidal ideation than their male counterparts. The result of this study showed that 50% of female Korean youth reported some level of suicidal ideation, while 36% of male Korean youth reported some level of suicidal ideation. In addition, female youth reported greater levels of depression and social anxiety than males and additionally reported experiencing greater levels of all studied somatic complaints (i.e., problems with sleep, fatigue, nausea, breathing, fever, headaches, and loss of appetite) than males. These findings are consistent with previous research showing that females experience greater rates of internalizing problems (e.g., mood and anxiety disorders) than their male counterparts [48].

Additionally, our findings suggest that sleep issues serve as the strongest predictor for risk across all students, for both male and female groups. This finding is supported by previous research demonstrating the relation between sleep and suicide risk and that sleep disturbances tend to be a robust risk factor for suicidality [49,50,51]. Our findings are also in line with research specifically examining sleep and suicide risk in Korean youth. Several studies examining Korean youth populations have found similar results indicating that sleep disturbances are associated with an increased risk of suicidality [52,53,54] in adolescents. Therefore, mental health professionals are encouraged to consider and identify youth clients who are experiencing persistent and pervasive sleep issues and further consider suicidality risk.

Beyond the relationship between sleep and suicidal ideation, our findings also support previous research demonstrating the relation between somatic symptoms and suicidal ideation. For example, loss of appetite and feelings of nausea are associated with increased suicide risk in youth [38,55]. In addition, this study found that reporting headaches or fever is a useful predictor of risk in male students, while reporting nausea is a useful predictor of risk in female students. These findings are supported by previous studies, such as research by Jeon and his colleagues [56], which found a different relationship between suicidal ideation and somatic symptoms across male and female Korean outpatients. Therefore, the assessment of multiple somatic symptoms (i.e., sleep, headache, nausea) should be emphasized in order to distinguish the complex and diverse profiles of mental health across genders, with special attention paid to the strongest predictive somatic symptom (i.e., sleep).

While this study is aimed at helping mental health professionals to screen for suicidality, additional interventions are warranted. Within the United States, somatic symptomatology has begun to be integrated within a psychological evaluation. For example, the Patient Health Questionnaire (PHQ-9) is a self-administered assessment aimed at measuring depression [57] and was created to be implemented in primary healthcare settings. The PHQ-9 measure asks questions on physical symptomatology regarding appetite, sleep, and motor functioning. Utilizing measures that integrate somatic symptoms is necessary to fully encapsulate mental health symptomatology and would be beneficial given prior research suggesting that Korean populations are more likely to report somatic symptoms than cognitive or emotional problems on assessment measures [58].

In order to implement effective screening and prediction systems for suicide, it is necessary to consider cultural and contextual factors surrounding suicide in each society. For example, discourse regarding suicide in the United States focuses predominantly on individual pathology; however, discourse in Japan has focused on social pathology, economic factors, and the “culture of suicide” [59]. In 2003, Japan reported a suicide rate of 27 per 100,000, after experiencing an increase in suicide rates in the previous few decades. Following a gradual decrease beginning in 2009, the number of suicides in Japan fell below 30,000 in 2012 for the first time since 1998 [5]. This decrease is significant and is a direct response of Japan’s governmental and societal shift in mental health and suicide attitudes that began in 2006. Takeshima and authors [60] describe societal-level interventions such as prevention measures aimed at increasing public awareness and the revision and enactment of laws to support high-risk individuals (e.g., Act for the Prevention of Child Abuse; Money Lending Business Act revision). Public and governmental awareness led the Japanese government to allocate substantial funds to local government and mental health agencies to aid in suicide prevention measures [60]. However, during Japan’s second wave of the COVID-19 pandemic (July–October 2020), suicide rates increased by 16%. The groups with the highest increase in suicides were children and adolescents, with a 49% increase, which the authors hypothesized to be associated with the reopening of schools [61]. Although the mental health effects from the COVID-19 pandemic are still being measured, the previous literature demonstrates that youth suicide coincides with the academic school calendar [62,63]. Therefore, suicide screening and interventions are more efficacious when developed with an understanding of the social, as well as the individual, context. 

Although this research suggests a multidimensional picture of Korean youth mental health, it has a few limitations. First, only first-year middle school students were included in this study. Even though a nationally representative sample was used in this study, considering the different developmental contexts within adolescence, the results of this research are more difficult to generalize to the different age groups within the adolescent population. Additional research should aim to demonstrate whether somatic symptoms are useful in predicting at-risk groups using a broader sample across a wider age range. Second, suicidal ideation was measured using a single-item self-report measure. Single- item self-report measures are used frequently in larger-scale populations and nationally representative research [64] because they reduce the respondent and administrative burden [65]. Further, self-report measures are beneficial as they can assess internal experiences, such as depression and anxiety, which may not be readily observable utilizing other data collection methods. Self-report measures can also aid in reducing shame and stigma as they are typically de-identified, as is the case with the Korea Children and Youth Panel Survey. An example of a self-report survey is the Global School-Based Student Health Survey (GSHS). The GSHS is a nationally representative self-report survey created by the United Nations and UNICEF and administered to adolescents in various countries, utilizing single-item measures to examine various domains, such as suicidal ideation. While using a multiple-item measure for suicidal ideation is preferred, this evidence suggests that using a single item within the context of large-scale research is acceptable, although some research has demonstrated the low validity of single-item measures examining suicidality and suggests multi-step instruments for examining suicide in order to consider the diverse aspects and severity of suicidality [66]. Additionally, future research is recommended to consider other possible limitations of using self-report measures, such as the under-reporting of psychological and somatic symptoms or misinterpretation of questions by participants.

## 5. Conclusions

When initially treating Korean youth, mental health professionals are encouraged to identify and recognize somatic complaints as potentially representative of mental health concerns. In particular, sleep issues should be a critical factor when understanding Korean youth’s suicidal ideation, depression, and social anxiety. Assessing other somatic symptoms (e.g., headaches and nausea) is also important for understanding Korean youth mental health because doing so helps to distinguish different profiles across suicidal ideation, depression, and social anxiety. Evaluating and asking questions about somatic complaints would be useful to combat the high rates of stigma and low rates of disclosure experienced by Korean youth and aid mental health professionals in early identification and treatment.

## Figures and Tables

**Figure 1 behavsci-11-00151-f001:**
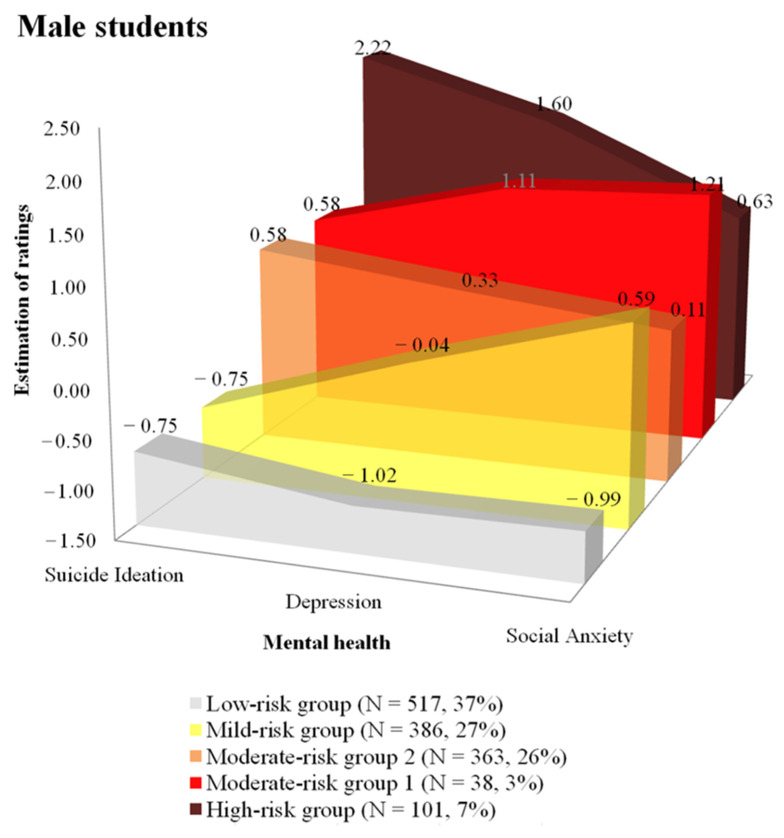
Five profiles in the mental health of Korean male youth.

**Figure 2 behavsci-11-00151-f002:**
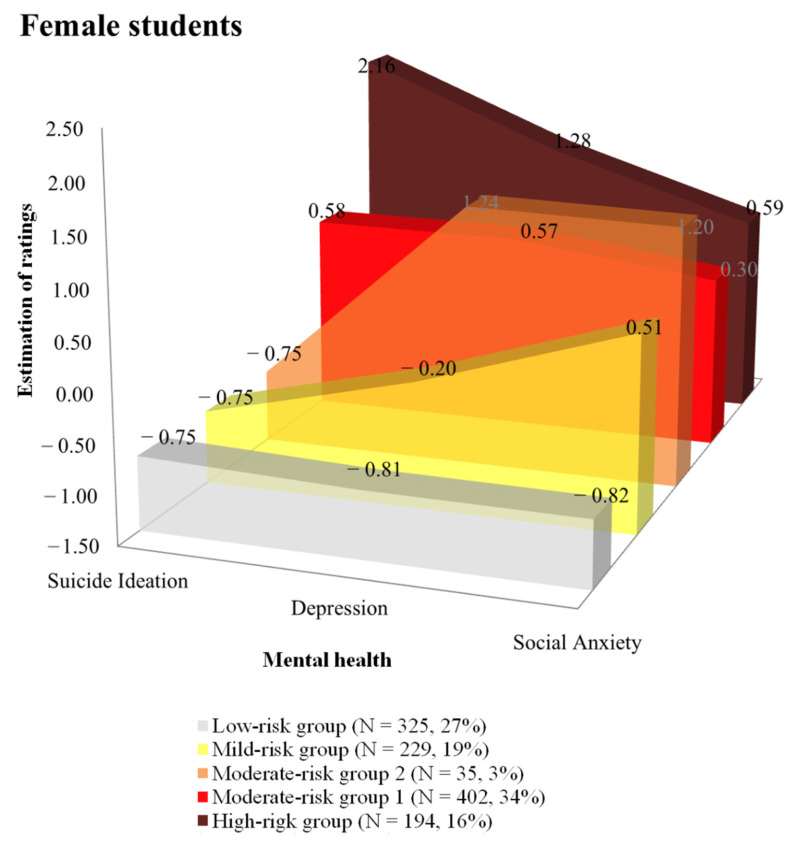
Five profiles in the mental health of Korean female youth.

**Figure 3 behavsci-11-00151-f003:**
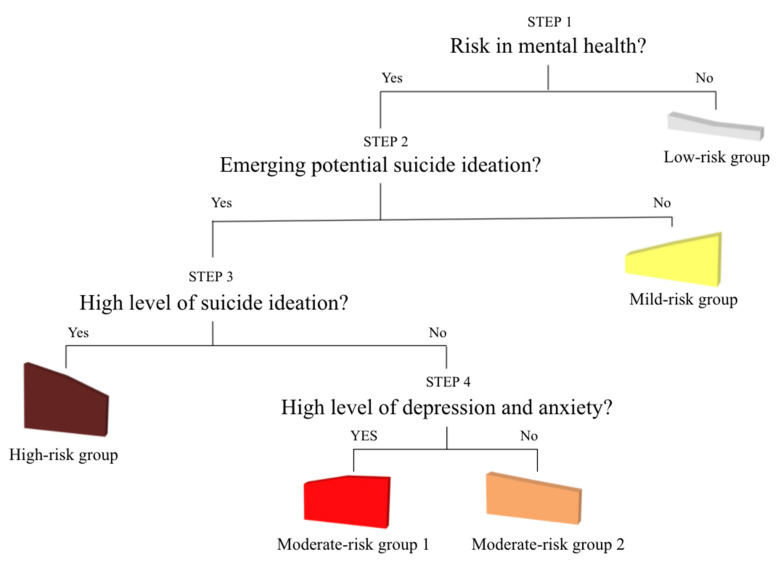
Screening steps across the latent profiles of male students.

**Figure 4 behavsci-11-00151-f004:**
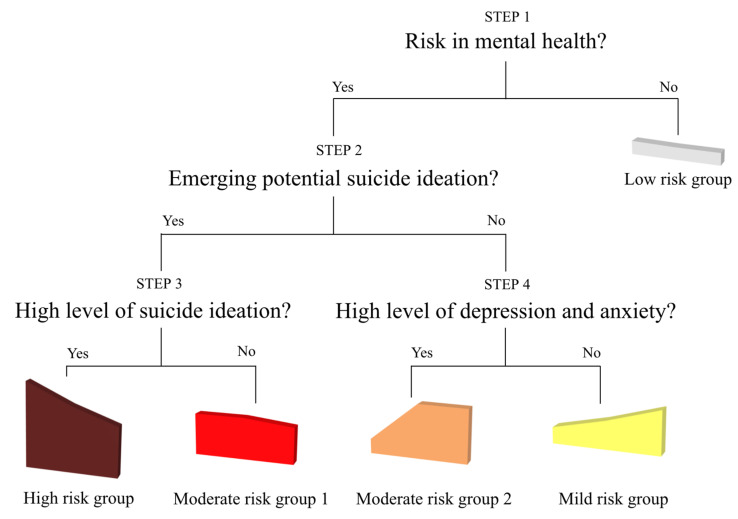
Screening steps across the latent profiles of female students.

**Table 1 behavsci-11-00151-t001:** Model fit information by the number of latent profiles.

Gender	Profiles	AIC	BIC	SSABIC	Entropy	Adj. LRT
Male	2	10,331.98	10,384.46	10,352.69	0.95	-
	3	7768.593	7842.062	7797.589	0.99	0.049
	4	7313.878	7408.338	7351.159	0.91	0.000
	5	7263.248	7378.700	7308.814	0.90	0.001
	6	7200.809	7337.252	7254.660	0.87	0.167
Female	2	9517.120	9567.895	9536.131	0.75	-
	3	8077.246	8148.331	8103.862	0.99	0.015
	4	7904.613	7996.008	7938.833	0.91	0.000
	5	7882.058	7993.763	7923.883	0.87	0.007
	6	7862.872	7994.887	7912.302	0.84	0.075

**Table 2 behavsci-11-00151-t002:** Logistic regression with somatic symptoms.

		Step 1				Step 2				Step 3				Step 4			
		*B*	*S.E.*	OR	95% CI	*B*	*S.E.*	OR	95% CI	*B*	*S.E.*	OR	95% CI	*B*	*S.E.*	OR	95% CI
Male	Constant	1.22 ***	0.09	3.37		0.07	0.07	1.07		−1.80 ***	0.16	0.17		−4.19 ***	0.44	0.02	
	Sleep	0.25 **	0.09	1.28	[1.06, 1.54]	0.18 *	0.09	1.19	[1.00, 1.42]	0.11	0.14	1.12	[0.86, 1.47]	0.63 **	0.24	1.87	[1.17, 2.98]
	Headache	0.08	0.10	1.08	[0.90, 1.31]	0.11	0.1	1.12	[0.92, 1.37]	0.27	0.17	1.31	[0.94, 1.84]	0.92 **	0.29	2.5	[1.41, 4.45]
	Nausea	0.19	0.11	1.20	[0.97, 1.49]	0.18	0.10	1.2	[0.98, 1.47]	0.07	0.16	1.07	[0.78, 1.47]	−0.16	0.3	0.85	[0.48, 1.53]
	Stomach Ache	0.21 *	0.10	1.23	[1.01, 1.50]	0.23 *	0.09	1.26	[1.05, 1.51]	0.19	0.15	1.20	[0.90,1.60]	0.24	0.26	1.27	[0.76, 2.13]
	Appetite	0.28 **	0.09	1.32	[1.11, 1.56]	0.06	0.09	1.06	[0.89, 1.28]	0.25	0.16	1.28	[0.93, 1.75]	0.44	0.3	1.55	[0.87, 2.78]
	Tiredness	0.37 ***	0.08	1.45	[1.23, 1.71]	−0.07	0.09	0.93	[0.77, 1.12]	0.07	0.17	1.08	[0.77, 1.50]	0.98 **	0.32	2.68	[1.44, 4.97]
	Breathing	0.43 ***	0.11	1.53	[1.25, 1.89]	0.22 *	0.09	1.25	[1.04, 1.49]	0.02	0.15	1.02	[0.77, 1.35]	0.07	0.25	1.07	[0.66, 1.73]
	Fever	0.86 ***	0.13	2.37	[1.85, 3.02]	0.05	0.10	1.05	[0.87, 1.27]	−0.04	0.16	0.96	[0.69, 1.32]	0.28	0.27	1.32	[0.78, 2.22]
		**Step 1**				**Step 2**				**Step 3**				**Step 4**			
		** *B* **	** *S.E.* **	**OR**	**95% CI**	** *B* **	** *S.E.* **	**OR**	**95%CI**	** *B* **	** *S.E.* **	**OR**	**95%CI**	** *B* **	** *S.E.* **	**OR**	**95%CI**
Female	Constant	1.23	0.09	3.42		0.62	0.08	1.86		−1.21	0.12	0.30		−2.35	0.25	0.1	
	Sleep	0.36 **	0.10	1.43	[1.17, 1.75]	0.28 **	0.09	1.33	[1.11, 1.59]	0.23 *	0.10	1.26	[1.04, 1.54]	0.00	0.22	1	[0.65, 1.52]
	Headache	−0.08	0.11	0.92	[0.75, 1.14]	0.16	0.11	1.17	[0.94, 1.45]	0.16	0.13	1.18	[0.91, 1.52]	−0.05	0.26	0.95	[0.57, 1.58]
	Nausea	0.37 **	0.12	1.44	[1.14, 1.82]	−0.08	0.11	0.92	[0.75, 1.14]	0.03	0.12	1.03	[0.81, 1.32]	0.24	0.25	1.27	[0.77, 2.09]
	Stomach Ache	0.25 **	0.11	1.28	[1.04, 1.58]	0.05	0.09	1.05	[0.87, 1.26]	−0.02	0.10	0.98	[0.80, 1.20]	0.55 **	0.19	1.74	[1.19, 2.54]
	Appetite	0.30 **	0.09	1.35	[1.12, 1.61]	0.09	0.09	1.1	[0.92, 1.31]	−0.11	0.11	0.89	[0.72, 1.11]	0.05	0.23	1.06	[0.67, 1.66]
	Tiredness	0.3 **	0.09	1.35	[1.14, 1.60]	0.05	0.09	1.05	[0.87, 1.27]	0.30 *	0.12	1.34	[1.06, 1.70]	0.36	0.26	1.43	[0.86, 2.39]
	Breathing	0.38 **	0.12	1.46	[1.16, 1.83]	0.26 **	0.10	1.29	[1.07, 1.56]	0.09	0.11	1.10	[0.89, 1.35]	0.25	0.20	1.28	[0.87, 1.90]
	Fever	0.22	0.12	1.24	[0.99, 1.56]	0.11	0.10	1.11	[0.91, 1.37]	0.23	0.12	1.26	[0.99, 1.59]	−0.03	0.23	0.97	[0.61, 1.54]

* *p* < 0.05, ** *p* < 0.01, *** *p* > 0.001.

## Data Availability

The data are openly accessible through the website of the National Youth Policy Institute (www.nypi.re.kr) (11 November 2020).

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
