# Peer review of "Understanding Somatic Symptoms Associated with South Korean Adolescent Suicidal Ideation, Depression, and Social Anxiety"

_behavsci, 2021, doi:10.3390/bs11110151_

Round 1

Reviewer 1 Report

Thank you for the opportunity to review this manuscript. The authors conducted a latent profile analysis using the freely available KCYPS 2018 dataset that included 2.590 South Korean youths. Dependent variables included suicidal ideation, depression, and social anxiety. Independent variables included sleep, tiredness, appetite, nausea, breathing, fever, and headache. Five-profile models from male and female students were considered best for describing the profiles across dependent variables of each group. The authors recommend mental health professionals identify somatic complaints as possible indicators of mental health problems and increased risk of suicide. Overall, I think the authors did a really nice job in presenting their rationale for the study, clearly reporting the results, and discussing implications.

I only have the following suggestions for the authors:

  • Although the authors acknowledge the limitation of having a measure of suicide ideation that includes only one item, this seems to be a significant issue that warrants further justification.
  • Specify the software package used for the analysis.

I believe this study would be of interest to readers and recommend publication after minor revisions.

Author Response

Reviewer 1

No

Comment

Response

1

Although the authors acknowledge the limitation of having a measure of suicide ideation that includes only one item, this seems to be a significant issue that warrants further justification.

We added supporting literature regarding the use of single-item measures, especially in the context of larger-scale population based data, in the limitation section of the discussion, located on page 14. 

2

Specify the software package used for the analysis.

Mplus is the software used in this study. Unlike R or Python, Mplus doesn’t use individual packages for each analysis. However, we added the version and citation of the software.

Reviewer 2 Report

Solberg et al present an interesting and important study into the association between somatic symptoms and suicidal ideation in South Korean adolescents. The introduction is very well-written and provides a thorough background and compelling argument. I do think overall it is a good study. However, there are a number of methodological considerations that would greatly enhance the impact of the paper.

  1. I think the group names are confusing, I would suggest a more concise/consistent naming convention. Also “emerging” sounds like you are aware these symptoms are new, but you do not have this information. It would be more accurate to describe them as mild, moderate, severe, perhaps something like the follows (for males):

High risk group – high suicide ideation, depression and anxiety

Moderate risk group 1 – moderate suicide ideation, depression and anxiety

Moderate risk group 2 - Moderate suicide ideation, mild depression and anxiety

Mild risk group - Low suicide ideation and depression, mild anxiety

Low risk group – Low suicide ideation, depression and anxiety

Adopting a more consistent naming convention will also help with interpretation and explanation of the first paragraph of results section Somatic symptoms associated with mental health profiles which is currently very confusing.  

  1. There appears to be an analysis between high risk groups vs. low risk groups, but nowhere is it defined what categories fit into the high risk vs. low risk groups?
  2. I acknowledge that the ‘screening steps’ are post hoc and based on the development of the latent classes, however it is unclear where the methodological rational for this process came from. How and why did investigators develop these ‘steps’? Although it is a post hoc analysis, the rational for the method chosen should have been developed a priori and should be included in the methods section as opposed to the results.
  3. Why did authors choose to do logistic regression as opposed to describing/comparing the different categories in terms of their somatic symptoms? In my mind this is probably equally as informative but not as confusing and with less potential for spurious results. Further, the ‘step’ process assumes that step 1 – step 4 is a progression, but this is not necessarily true and the secondary data used for this analysis is not fit for purpose in terms of its ability to investigate this. Just because suicidal ideation is milder in one group that another doesn’t mean that group has ‘emerging’ symptoms. These symptoms may have been present for years. They do, however, have more severe symptoms.
  4. I think the initial male versus female analysis is unnecessary. There is enough literature to support stratifying the analyses by sex without the need for the MANOVA. You should state from the beginning the analyses were performed separately for males and females and cite the relevant literature that males and females have very different patterns in terms of mental health/somatic symptoms

Minor comments:

I am unsure what the following sentence means: “The standardized values were used for each of the dependent variables.”

None-risk is mentioned in the discussion, typo?

Given the compelling introduction with regards to shame and stigma associated with reporting of poor mental health in Korean youth, what limitations in terms of the self-report questionnaire are considered by the authors?

I like Fig 1 and 2, but the axes need more explanation.

In the following sentence: “That is, if the p-value of class N is higher than 0.05, it means the model with N profiles is not significantly different from the model with N profiles, and it is recommended to select a model with N-1 profiles rather than N profiles.” Should this be: “That is, if the p-value of class N is higher than 0.05, it means the model with N profiles is not significantly different from the model with N-1 profiles, and it is recommended to select a model with N-1 profiles rather than N profiles.”

Author Response

Reviewer 2

No

Comment

Response

1

I think the group names are confusing, I would suggest a more concise/consistent naming convention. Also “emerging” sounds like you are aware these symptoms are new, but you do not have this information. It would be more accurate to describe them as mild, moderate, severe, perhaps something like the follows (for males): 

High risk group – high suicide ideation, depression and anxiety 

Moderate risk group 1 – moderate suicide ideation, depression and anxiety 

Moderate risk group 2 - Moderate suicide ideation, mild depression and anxiety 

Mild risk group - Low suicide ideation and depression, mild anxiety 

Low risk group – Low suicide ideation, depression and anxiety 

Adopting a more consistent naming convention will also help with interpretation and explanation of the first paragraph of results section “Somatic symptoms associated with mental health profiles” which is currently very confusing.

We re-named groups like below in the paragraphs and figures.  

Male groups

High risk group

Moderate risk group 1

Moderate risk group 2

Mild risk group

Low risk group

Female groups

High risk group

Moderate risk group 1

Moderate risk group 2

Mild risk group

Low risk group

2

There appears to be an analysis between high risk groups vs. low risk groups, but nowhere is it defined what categories fit into the high risk vs. low risk groups?

To define what categories fit into each group, Latent Profile Analysis was used in this research. Additional explanations of the method was added on page 4.

3

I acknowledge that the ‘screening steps’ are post hoc and based on the development of the latent classes, however it is unclear where the methodological rationale for this process came from. How and why did investigators develop these ‘steps’? Although it is a post hoc analysis, the rationale for the method chosen should have been developed a priori and should be included in the methods section as opposed to the results.

In this study, the post hoc process could only be planned after receiving the results of the Latent Profile Analysis (LPA). So, the post hoc process was described in the results rather than methods to prevent the confusion of readers with the latent groups. However, we added more details for the methodological rationale of the post hoc analysis on page 5.

4

Why did authors choose to do logistic regression as opposed to describing/comparing the different categories in terms of their somatic symptoms? In my mind this is probably equally as informative but not as confusing and with less potential for spurious results. Further, the ‘step’ process assumes that step 1 – step 4 is a progression, but this is not necessarily true and the secondary data used for this analysis is not fit for purpose in terms of its ability to investigate this. Just because suicidal ideation is milder in one group that another doesn’t mean that group has ‘emerging’ symptoms. These symptoms may have been present for years. They do, however, have more severe symptoms.

We discussed the rationale of choosing logistic regression in the method (page 5). The step process was not necessary designed with the assumption of progression. The step is comparison process between different groups so that this research’s finding can provide an example of practical screening strategies. However, we agree the term “emerging” may not fit well to describe mild level group because “emerging” can imply the prevalence of symptoms. So, we changed the group’s name from “emerging” to  “moderate” or “mild” to prevent confusion.

5

I think the initial male versus female analysis is unnecessary. There is enough literature to support stratifying the analyses by sex without the need for the MANOVA. You should state from the beginning the analyses were performed separately for males and females and cite the relevant literature that males and females have very different patterns in terms of mental health/somatic symptoms

We added more literature on gender differences in the introduction. While the literature indicates higher suicidal ideation rates for women, we have separated the analysis by Gender because our questions also involve whether and to what extent the patterns of symptoms were similar or different. The unique pattern for males and females can be understood within the context of development, with female youth experiencing pubertal maturity, which in turn provokes different physiological and psychological changes than males, therefore influencing reported mental and somatic complaints. For example, our results also indicated that males were found to have three patterns (groups) that included suicidal ideation while female suicidal ideation was found in two groups.

6

I am unsure what the following sentence means: “The standardized values were used for each of the dependent variables.”

We expanded and reworded this sentence- it now says “The standardized values for each independent variable were used for predicting the dependent variables.”

7

None-risk is mentioned in the discussion, typo?

Yes, this was a typo. It now says “low-risk”. 

8

Given the compelling introduction with regards to shame and stigma associated with reporting of poor mental health in Korean youth, what limitations in terms of the self-report questionnaire are considered by the authors?

We understand there might be limitations to using a self-report questionnaire. However, in nationally representative sample research it has been commonly used because it reduces respondent and administrative burden (we added supporting literature for this on page 14), especially when it is an anonymous survey. In the introduction, we especially wanted to discuss how the cultural context impacts disclosure to their peers, parents, communities.

9

I like Fig 1 and 2, but the axes need more explanation.

The labelings of axes were added in Fig 1 and 2.

In the following sentence: “That is, if the p-value of class N is higher than 0.05, it means the model with N profiles is not significantly different from the model with N profiles, and it is recommended to select a model with N-1 profiles rather than N profiles.” Should this be: “That is, if the p-value of class N is higher than 0.05, it means the model with N profiles is not significantly different from the model with N-1 profiles, and it is recommended to select a model with N-1 profiles rather than N profiles.”

We changed the sentence following the comment. We appreciated for detailed and important comment. 

Round 2

Reviewer 2 Report

The manuscript is much improved. Only a couple of minor comments. Firstly, I limitations with regard to self-report bias need to be mentioned. Secondly, althoug the group names have changed in text all the figures have the old group names. Can these please be changed to match the text?

Author Response

Comment

Response

Firstly, I limitations with regard to self-report bias need to be mentioned. 

Self-report bias has been added to the limitation section of the discussion

Secondly, althoug the group names have changed in text all the figures have the old group names. Can these please be changed to match the text?

All figures were replaced with the new group names.
